# True Diaphragmatic Hernia (Morgagni Hernia) Incidentally Diagnosed with Positive Contrast Peritoneography in a Cat: A Rare Case Report and a Review

**DOI:** 10.3390/vetsci11040159

**Published:** 2024-04-01

**Authors:** Jack-Yves Deschamps, Théo Corbarieu, Nour Abboud, Françoise A. Roux

**Affiliations:** Emergency and Critical Care Unit, Oniris VetAgro Bio, Nantes-Atlantic College of Veterinary Medicine, Food Science and Engineering, La Chantrerie, CS 40706, 44 307 Nantes, France; theo.corbarieu@oniris-nantes.fr (T.C.); nour.abboud@oniris-nantes.fr (N.A.); francoise.roux@oniris-nantes.fr (F.A.R.)

**Keywords:** diaphragmatic hernia, true diaphragmatic hernia, Morgagni hernia, peritoneography, incidentaloma, cat

## Abstract

**Simple Summary:**

An 18-month-old domestic shorthair cat survived a fall from the second floor with minor injuries and no breathing issues. Routine radiographs suggested an unusual diaphragmatic hernia, which, contrast peritoneography revealed to be a pleuroperitoneal hernia—a congenital defect. This was confirmed and surgically repaired during laparotomy. The cat fully recovered. This case emphasizes the diagnostic value of peritoneography for hernias. The diaphragmatic defect, being ventral and to the right, corresponds to the right anterolateral hernia found in humans, known as “Morgagni hernia”.

**Abstract:**

An 18-month-old neutered male domestic shorthair cat was presented for an emergency consultation after falling from the second floor. The cat sustained minor traumatic injuries but did not exhibit dyspnea. Routine radiographic examination raised suspicion of a diaphragmatic hernia, but the circumscribed nature of the soft tissues visible in the thorax was atypical for a classic traumatic diaphragmatic hernia. A positive contrast peritoneography highlighted the likely presence of a hernial sac, which strongly suggested a “true diaphragmatic hernia”, also known as “pleuroperitoneal hernia”. This diagnosis was confirmed during laparotomy, which allowed for the visualization of a 3 cm radial diaphragmatic defect in the right ventral quadrant of the pars sternalis. The diaphragm’s edges were rounded. A portion of the falciform ligament and a part of the omentum were protruding through the defect and were contained within a hernial sac. Herniorrhaphy was performed. The cat recovered without complications. Given its presentation and location, ventrally and to the right, this anomaly is analogous to what is described in humans as “Morgagni hernia”. Six other cases of Morgagni hernias have probably been reported in cats but were not identified as such. This case underscores the utility of peritoneography, a straightforward technique useful for diagnosing diaphragmatic hernias, which enables differentiation between acquired traumatic forms and congenital forms, particularly peritoneopericardial hernias and pleuroperitoneal hernias. True diaphragmatic hernias are almost always serendipitous discoveries.

## 1. Introduction

Diaphragmatic hernias in cats are most often the result of trauma. These classic diaphragmatic hernias are not, in the strictest sense, real hernias. Indeed, a hernia presupposes the existence of a hernial sac. This is the case for umbilical hernias, inguinal hernias, and perineal hernias where the hernial sac is formed by the peritoneum.

A traumatic diaphragmatic hernia is an acquired opening of the diaphragm after which the abdominal organs move into the thoracic cavity (which has lower pressure) but are not enclosed in a hernial sac. The use of the term “hernia” in this context is incorrect; traumatic diaphragmatic hernias are false hernias, but given that traumatic diaphragmatic ruptures are much more common, it is customary to refer to them as “diaphragmatic hernias” and to specify the type of hernia when it does not correspond to the traumatic form.

Alongside these acquired forms, almost all of which are traumatic, there are congenital forms. Congenital diaphragmatic hernias arise from the incomplete development of a portion of the diaphragmatic dome. Among congenital diaphragmatic hernias, there are hernias occurring through the diaphragm’s natural openings: the esophageal hiatus and the foramen of the vena cava. Generally, these are not associated with a hernial sac. A few cases of hiatal hernias have been reported in cats [1,2,3,4,5,6,7] and only two cases of herniations through the foramen of the vena cava have been reported in cats [8,9].

Genuine diaphragmatic hernias—that is, with a hernial sac—are congenital. These are “peritoneopericardial hernias” (the most common in cats) and “pleuroperitoneal hernias”. Although peritoneopericardial hernia is a true hernia, it is customary to reserve the term “true diaphragmatic hernias” exclusively for pleuroperitoneal hernias, a convention that will be adhered to in the remainder of this article.

The definitive diagnosis of diaphragmatic hernias can be challenging. Positive contrast peritoneography can assist in identifying ambiguous cases. This paper reports an instance of an injured cat where this method serendipitously revealed an unexpected and rare presentation of a congenital diaphragmatic defect: a “true diaphragmatic hernia”, also called, “pleuroperitoneal hernia”. Other accounts of this uncommon congenital malformation in cats are reviewed.

## 2. Case Presentation

A one-and-a-half-year-old neutered male domestic shorthair cat was presented to the emergency department of our institution after falling from the second floor (five meters) 5 h earlier. No other issues, including trauma, were reported in the medical history prior to this event.

### 2.1. Physical Examination

On admission, the cat weighed 5.7 kg, had a body condition score of 6/9, a rectal temperature of 38.2 °C, a heart rate of 176 beats per minute, and exhibited tachypnea with 44 breaths per minute. The femoral pulse was strong and synchronous with the precordial activity. The oral mucous membranes were pink and moist, and capillary refill time was less than two seconds. A right sublingual ranula was observed, along with a degloving injury on the third and fourth digits of the right hind limb.

### 2.2. Additional Tests

A routine trauma assessment was carried out, including blood tests (hematocrit, proteinemia, glycemia, uremia, creatininemia, lactatemia, blood gases, ionogram), which were all within the range values. A systolic blood pressure measurement revealed slight hypertension (156 mmHg), which was attributed to stress or pain.

Plain chest radiographs identified a large radiodense area (4 cm × 4 cm), well circumscribed, homogeneous, with a density similar to soft tissue, in the usual projection area of the thorax. This radiodense area was located caudally and ventrally on the lateral projection, medially and just right to the spine on the dorsoventral projection, in contact with the sternum, the diaphragm and the apex of the heart, ventrally to the caudal vena cava (Figure 1a,b). The rest of the thoracic cavity appeared unremarkable. This finding, in the context of a fall from the second floor, raised the possibility of abdominal organs being implicated in a traumatic diaphragmatic rupture. The differential diagnosis included an incidentally discovered mass (granuloma, hematoma, abscess, cyst, neoplasia), a longstanding traumatic diaphragmatic hernia, or a congenital diaphragmatic hernia.

Among congenital diaphragmatic hernias, the very ventral location excluded a hiatal hernia. On both lateral and frontal radiographs, the radiodense area did not appear to be in continuity with the pericardial sac: the contours of the heart were clearly observable and the edges of the pericardial sac were clearly separated from this radiodense area, which did not support the diagnosis of a peritoneopericardial hernia. Hence, a pleuroperitoneal hernia was deemed the most likely diagnosis.

An abdominal and thoracic ultrasound exam failed to definitively confirm the presence of a diaphragmatic tear and to determine the mass’s composition.

The following day, a positive contrast peritoneography was carried out without sedation by injecting transabdominally, on the linea alba, 2 mL/kg of an iodine derivative, iopamidol (Iopamiron^®^, 300 mg/mL), at room temperature, and then placing the animal in sternal recumbency with the pelvis elevated (Figure 2a,b).

The contrast medium distributed well throughout the peritoneal cavity and was also observed in the usual thoracic projection area. However, it did not diffuse throughout the entire thoracic cavity; rather, it appeared to be contained by a membrane or by the deformed diaphragmatic muscle. The injected contrast agent, now present at the site of the abnormal structure previously seen in plain radiographs, confirmed the abdominal origin of this structure. Given the location, it could have been the falciform ligament, a part of a hepatic lobe, the omentum, the intestine, the stomach, the pancreas or the spleen.

### 2.3. Diagnosis

These findings led to the diagnosis of a diaphragmatic hernia. The very circumscribed nature of the hernia, and particularly the fact that the contrast medium did not disperse throughout the entire thoracic cavity, suggested a specific type of congenital diaphragmatic hernia known as “pleuroperitoneal hernia” or “true diaphragmatic hernia”, in which the involved organs are enclosed within a hernial sac. The hypothesis of a peritoneopericardial hernia was ruled out because, in such a case, the contrast medium would have spread into the pericardial sac. Similarly, had there been a diaphragmatic tear, the contrast medium would have disseminated throughout the pleural cavity. This case was deemed an incidental discovery of a congenital anomaly predating the traumatic event.

### 2.4. Surgical Treatment

In the wake of peritoneography, although the cat did not present any dyspnea, surgical intervention was conducted to prevent the involvement of additional organs, organ strangulation, and the onset of breathing difficulties.

The cat was anesthetized with an association of diazepam (Diazépam TVM^®^, 0.25 mg/kg IV), propofol (Propovet^®^, 4 mg/kg IV), and morphine (Morphine Aguettant^®^; 0.05 mg/kg) administered intravenously, then maintained with isoflurane (Vetflurane^®^, 1.5%). A midline laparotomy extending from the xiphoid process to the umbilicus did not reveal any muscular damage or bleeding that would indicate trauma at this location. The laparotomy enabled the visualization of a 3 cm radial defect in the diaphragm, located in the right ventral quadrant of the pars sternalis. The diaphragm’s edges were rounded, which implied that the defect was longstanding in nature. These findings supported the diagnosis of a congenital rather than a recent traumatic origin for the diaphragmatic breach.

A part of the falciform ligament and a part of the omentum were engaged in the diaphragmatic defect. The organs involved were encapsulated within a hernial sac composed of a serous membrane. These findings confirmed the diagnosis of a pleuroperitoneal hernia and accounted for the abnormal tissue density seen in the thoracic radiographs. Once the hernia was reduced, the herniated tissues were found to be unharmed.

The edges of the diaphragm were revived by scissors section to promote the healing of the margins. Herniorrhaphy was performed using a simple continuous closure with absorbable braided suture (Vicryl^®^ 3/0, Ethicon, Johnson & Johnson international). The hernial sac was not removed. At the end of the procedure, transdiaphragmatic aspiration made it possible to collect several milliliters of air. It was not considered necessary to place a chest tube.

### 2.5. Postoperative Phase

There were no intraoperative or postoperative complications. Immediate postoperative radiographs confirmed hernia reduction and showed the persistence of a mild pneumothorax (Figure 3a,b). The hernial sac, which was not removed, was clearly visible on the lateral view (white arrows), it contained a small amount of air. The contrast medium was drained by the parietal and visceral peritoneum, then eliminated by the kidneys, as it was subsequently visible in the bladder.

The cat fully recovered and was discharged the day following surgery. Upon removal of the sutures twelve days after surgery, the cat was in perfect health: there was no dyspnea, the ranula had disappeared, and the skin lesions on the fingers were healed. Contacted by phone one year after the surgical intervention, the owners reported no health concerns.

## 3. Discussion

### 3.1. True Diaphragmatic Hernia

“True diaphragmatic hernias”, also known as “pleuroperitoneal hernias”, arise from a congenital anomaly of the diaphragm, which features an incomplete rupture and persistent serosa membrane on the diaphragm’s thoracic side, preventing open communication between the thoracic and abdominal cavities. Owing to the lower pressure within the thoracic cavity, abdominal organs may protrude into this space; however, they are encapsulated by this serosa, forming the hernial sac.

True diaphragmatic hernias are infrequently documented in the veterinary literature. We have cataloged 12 individual reports in cats [10,11,12,13,14,15,16,17,18,19,20] and 4 individual reports in dogs [21,22,23]. To our knowledge, no cases have been reported in a case series. Cases may have been reported in textbooks. Major congenital diaphragmatic defects observed in cats [24,25] or in young dogs [26,27,28,29] were not included in this review.

### 3.2. Location of the Congenital Hernias

In humans, there are two types of congenital diaphragmatic hernias linked to incomplete fusion of the diaphragm: hernias located in the posterolateral region, right or left, called “Bochdalek hernias” (first described in 1848 by Vincent Alexander Bochdalek) and hernias located in the anterior and retrosternal region, called “Morgagni hernias” on the right (first described by Giovanni Battista Morgagni in 1769) or “Larrey hernias” on the left (attributed to Dominique Jean Larrey in 1812). Some authors do not distinguish between right and left anterior hernias and describe all anterior congenital hernias as “Morgagni hernia” [30]; 90% of Morgagni hernias occur on the right side due to the pericardial attachments to the diaphragm that provide protection and support to the left side [31].

In the veterinary literature, the type of true diaphragmatic hernia is not specified by the authors. Vosges et al. [11] and Carriou et al. [15] drew a parallel between the true diaphragmatic hernia they observed in cats and the Bochdalek hernia described in humans. However, Bochdalek hernia is a posterolateral hernia [32], whereas the hernias described by these authors were ventral (anterior).

In the current case, the diaphragmatic defect, being radial, ventral, and to the right, corresponds to the right anterolateral hernia found in humans, known as “Morgagni hernia”. To the best of our knowledge, this is the first instance where the term “Morgagni hernia” is employed in veterinary medicine to describe this congenital anomaly. Nevertheless, six cases previously reported in cats [11,12,15,16,19,20]—including the cases by Vosges et al. [11] and Carriou et al. [15]—along with three cases in dogs [21,22] appear to be consistent with a Morgagni hernia, as the radiological images are directly comparable to those of the current case.

Only the case described by Green et al. [13], which reports radiopaque structures ventrally and in the left hemithorax, seems to align with the “Larrey hernia” described in humans. Among the four cases of true diaphragmatic hernias reported in dogs, only the case of Devereux et al. [23] concerns a small discontinuity of the ventral left diaphragm and can therefore be considered as a Larrey hernia.

In the three cases reported by Mann et al. [10], White et al. [14], and Rose et al. [18], the authors noted a radiopaque soft-tissue mass in the midportion of the diaphragmatic crus: unlike Morgagni or Larrey hernias, the mass was not ventrally positioned. This location seems to correspond more with a “caval foramen hernia”, even though the first two cases mention the presence of a hernial sac, which is unusual in humans.

The two cases published by Gombač et al. [17] are necropsy findings that do not provide enough detail to determine the precise location of the hernia.

In cats, all true diaphragmatic hernias are ventral, either right or left; there have been no reports of hernias that correspond to the Bochdalek hernia seen in humans (posterolateral) (Figure 4). In contrast, in humans, Morgagni and Larrey hernias account for only 2 to 5% of congenital diaphragmatic hernias and Bochdalek hernias represent at least 95% of congenital diaphragmatic hernias [30,31].

### 3.3. Serendipitous Discoveries

Most acquired diaphragmatic hernias occur post-trauma, typically due to a road accident or defenestration. In the present case, a fall from the second floor along with concurrent injuries (ranula and degloving of two digits) suggested a traumatic origin. However, what was initially supposed to be a traumatic diaphragmatic hernia was actually a congenital defect that was fortuitously revealed.

The cases reported in the veterinary literature are all serendipitous discoveries during the investigation of symptoms not typically associated with diaphragmatic hernias: intermittent diarrhea [11], hyperthyroidism [12], cough [14], acute neurological deterioration [15], squamous cell carcinoma of the ears [13], diabetes mellitus [18], routine health inspection [19], and intermittent vomiting [20].

In one case, a hernia was discovered while assessing injuries from a recent road traffic accident, yet the cat displayed no respiratory distress [16]; similar to our report, the hernia was considered unrelated to the recent trauma.

While the anomaly is congenital, nearly all reported cases (10 out of 13) involved adult cats, ranging from 15 months to 14 years old. Only three symptomatic cases were documented in young cats: one by Mann et al., which involved an eight-month-old cat that presented with respiratory distress [10], and two by Gombač et al. [17], regarding closely related British shorthair cats, where both succumbed postoperatively to severe dyspnea. Necropsy revealed a thin, flaccid, and distended transparent tendinous portion of the diaphragm protruding cranially into the thoracic cavity, forming a cupola in which the left, right medial, and quadrate hepatic lobes were encased in both cats, and the stomach in one cat.

In human medicine, Morgagni hernias are also often incidental findings [30,33,34,35,36,37,38,39,40,41,42].

### 3.4. Herniated Organs

The organs typically involved include a small portion of the liver in 10 out of 13 cases (77%) [10,12,14,15,16,17,18,19,20], or the falciform ligament in 4 out of 13 cases (30%) [11,13,19], including the current case. In the case documented by Lee et al. [19], both the liver and the falciform fat were present; in the present report, a part of the falciform ligament and a part of the omentum were involved. In humans, the intrathoracic hernial sac often encompasses the omentum and the falciform ligament [33].

Given their distinct margins and thoracic location, it is understandable that these hernias are sometimes misidentified as pulmonary masses [11,14,19].

### 3.5. Positive Contrast Peritoneography

When digestive loops containing air are involved, radiological and ultrasound diagnoses of diaphragmatic hernia are typically straightforward. However, when only the liver or a limited number of non-digestive organs are involved (such as the falciform ligament, the omentum or the spleen), diagnosing diaphragmatic hernia can be challenging. Misdiagnosis with pulmonary atelectasis is possible [14]. Barium transit is diagnostic only when the digestive tract is implicated.

Peritoneography is a somewhat neglected examination that can reveal a diaphragmatic breach when an iodinated contrast medium is seen in the thoracic cavity [43,44]. It also helps to clarify the hernia’s nature by indicating the presence of a hernial sac, as evidenced by the distinctly circumscribed nature of the contrast medium within the thoracic projection area. Since the contrast medium did not spread into the pericardial sac, this procedure ruled out a peritoneopericardial hernia. Peritoneography enabled the confirmation of the diagnosis of a true diaphragmatic hernia in one cat [16] and two dogs [21]; the images acquired from theses cases are fully comparable to those of the case we are presenting.

Some false negatives may occur with this procedure. In one instance, positive contrast peritoneography was attempted but failed to demonstrate any connection between the pleural and peritoneal cavities due to the sequestration of the contrast medium within the falciform ligament [10]. A study involving 35 cats demonstrated comparable accuracy between ultrasound and peritoneography for diagnosing traumatic diaphragmatic hernias, with a 94% diagnostic rate (33/35) [45]. In two cases from this group (6%), the contrast medium did not disperse into the thorax; the authors blame adhesions as a potential cause. However, adhesions are uncommon in cases of diaphragmatic hernia, implying that it is more probable that the organs completely block the diaphragmatic opening.

Iopamidol, a nonionic water-soluble contrast medium, has been evaluated for its reactivity when used in peritoneal contexts, like peritoneography; it does not provoke significant peritoneal reactivity [46].

Peritoneography has been selected due to its simplicity, the obviation of sedation, and its widespread accessibility to veterinarians. Computed tomography (CT) or magnetic resonance imaging (MRI) are more sensitive methods for diagnosing diaphragmatic hernias and are the preferred diagnostic tools in human medicine. In feline cases, three reports of true diaphragmatic hernias were identified through computed tomography [13,18,19]. On CT images, a thin membrane connected with the diaphragm and creating a hernial sac can be seen [19].

### 3.6. Surgery

Despite the absence of clinical symptoms associated with its diaphragmatic hernia, surgery was elected for the cat. Surgical intervention is generally the preferred course of action when a traumatic diaphragmatic hernia is suspected to prevent the entrapment of additional organs, organ strangulation, and the onset of respiratory distress symptoms. However, the decision for surgery in this particular case could be seen as debatable since a true diaphragmatic hernia was confirmed via peritoneography, with the organs theoretically contained within a hernial sac and, thus, unlikely to massively protrude into the thoracic cavity.

Nearly all veterinary case reports (eight out of ten) advocate for surgical intervention, with laparotomy confirming the diagnosis of a true diaphragmatic hernia and allowing the correction of the defect. In two reports, the asymptomatic anomaly was not surgically addressed, which is a conceivable approach [12,19]. In human medicine, most Morgagni hernias are identified only after they have progressed [33]; considering the risk of incarceration, surgical repair is advised for all diagnosed Morgagni hernias [31,33].

## 4. Conclusions

True diaphragmatic hernias, also known as pleuroperitoneal hernias, are uncommon congenital anomalies in cats, often asymptomatic and discovered incidentally. The case presented here is analogous to the Morgagni hernia found in humans. It highlights the value of a straightforward radiographic test, peritoneography, not just for confirming the presence of a diaphragmatic hernia, but also for determining its nature, here congenital, in a scenario where a typical traumatic diaphragmatic hernia was anticipated. The management of this incidental finding with surgery is debatable.

## Figures and Tables

**Figure 1 vetsci-11-00159-f001:**
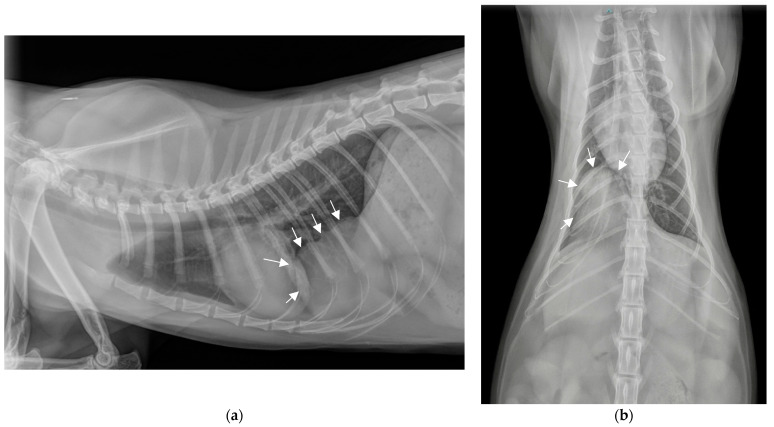
Chest radiographs on admission. (**a**) Profile view (latero-lateral) in right decubitus position. (**b**) Front view (ventro-dorsal).

**Figure 2 vetsci-11-00159-f002:**
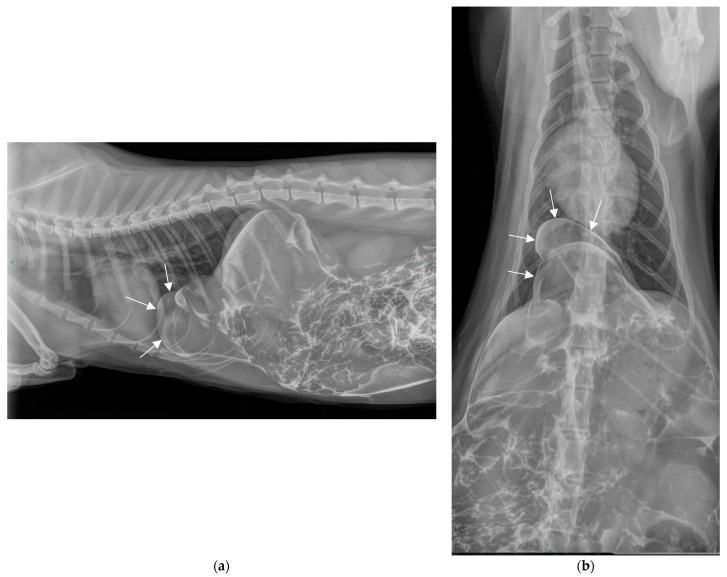
Thoracic and abdominal radiographs obtained 10 min after contrast medium injection. (**a**) Lateral view in right decubitus position. (**b**) Front view (ventro-dorsal).

**Figure 3 vetsci-11-00159-f003:**
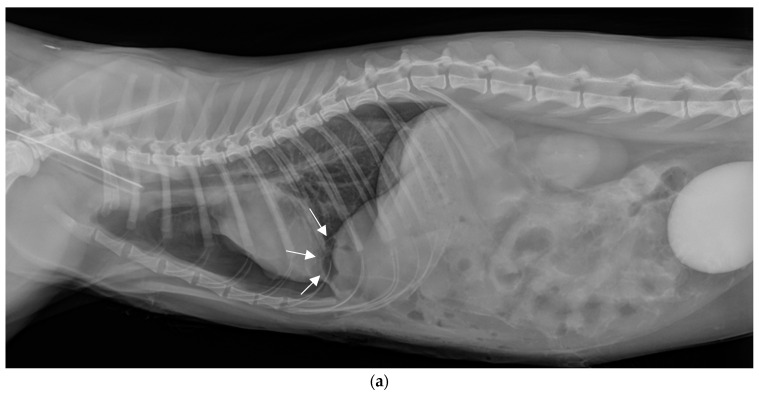
Thoracic and abdominal radiographs immediately postoperatively. (**a**) Lateral view in right decubitus position. (**b**) Front view (ventro-dorsal).

**Figure 4 vetsci-11-00159-f004:**
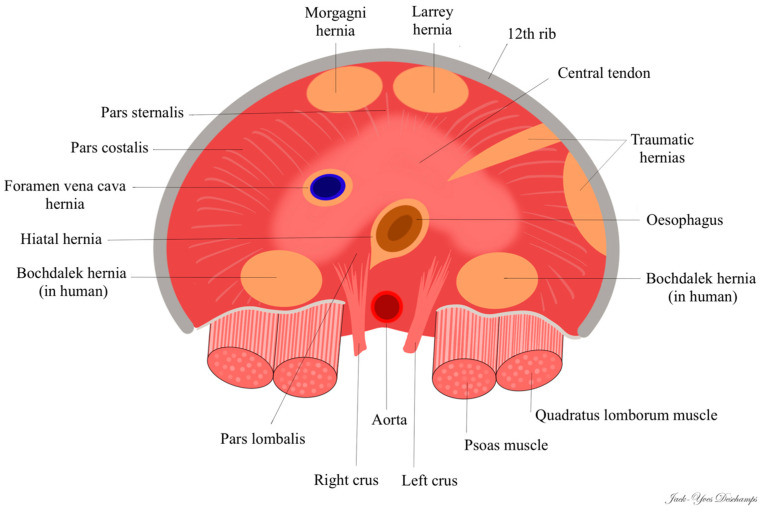
Location of diaphragmatic hernias in cats (view from the back, dorsal recumbency).

## Data Availability

All data are available on request.

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
