# Peer review of "True Diaphragmatic Hernia (Morgagni Hernia) Incidentally Diagnosed with Positive Contrast Peritoneography in a Cat: A Rare Case Report and a Review"

_vetsci, 2024, doi:10.3390/vetsci11040159_

Round 1

Reviewer 1 Report

Comments and Suggestions for Authors

I read the article with great interest, the study is interesting and can enrich our knowledge on the development of diaphragmatic hernias. But some clarifications need to be added before it can be considered for publication. 

Author Response

Dear reviewer, 

Thank you very much for the time you have devoted to reviewing this article .

In response to the feedback received, while I appreciate the call for additional clarifications, specific points needing further explanation were not identified. Therefore, I have focused on addressing the queries and concerns raised by the other reviewers. If you could provide more detailed guidance on the areas that require clarification, I would be more than willing to elaborate and improve the manuscript accordingly for consideration for publication.

Cordially

Reviewer 2 Report

Comments and Suggestions for Authors

The clinical case is well described and underlines the importance not only of a correct diagnosis but also of a correct diagnostic procedure on a pathology, in a general sense, currently found in clinical practice. The Authors very carefully highlight the specificities and differences between the different forms of this paratopia even if it brings nothing more than what is known.

Author Response

Dear reviewer, 

Thank you very much for the time you have devoted to reviewing this article and for the favorable impressions you have expressed.

Best regards

Reviewer 3 Report

Comments and Suggestions for Authors

The case report submitted by Deschamps et al. provides potentially relevant information regarding true diaphragmatic hernia in cats. Its main strength lies in the positive outcome.

However, the report requires some improvements to be accepted for publication.

The Introduction section is quite short, thus does not give a good insight into the topic. Some parts of the Discussion section should be moved to the Introduction.

The Case presentation section lacks information about the overall treatments performed to the cat, the time elapsed between events and the choices underlying certain treatments.

The Discussion section is interesting and clear but it could be perceived as redundant in some parts, as most of the concepts are repeated in different sections. The subdivision in many paragraphs is not so useful: the text could be structured as a single unit and written more concisely and with less redundancy. In this form, it seem a monography about the topic of hernia, rather than a discussion of the case description.
In this section, a part regarding the possible limitations of this patient’s management is lacking. There is no mention about the contraindication and adverse effects of the contrast peritoneography.

Specific comments are listed below (please note that this refers to the Version 1, since this review was made on it. The version 2 was uploaded later).

Major:

-       Line 2. Since the terminology “Morgagni hernia” has never been used in cats, as you report, it is hazardous to use it directly in the title. I suggest to leave the description in the Abstract and in the keywords, but to remove it from the title.

-       Line 15. The word classic is used throughout the text (line 138, 229, and others), most of the times written within the quotation marks (“classic”). Personally, I would avoid using this term throughout the text to refer to a traumatic hernia. Instead, I would simply use “traumatic hernia”. This makes the text more fluid and formal. I recommend making similar adjustments in subsequent instances where the term “classic” is used.

-       Line 32. The introduction section is too short. Please extend it, maybe moving here some parts from the Discussion section.

-       Line 57. The presence of a mass is not confirmed, as this is only a radiographic interpretation. Thus, using the term “mass” is not correct in this phase. I would modify to “radiodense area”. I would change it throughout the text (line 77, and others).

-       Line 75. “front view (dorso-ventral)”. As the cat did not exhibited dyspnea, why a dorso-ventral projection was chosen, instead of a ventro-dorsal? Or is it just an error in the text? The same for the next Figures (lines 91, 135).

-       Lines 76-89. Were all these procedures and the surgery performed on the same day?

-       Line 79. Insert the drug name followed by the manifacturer’s information within parentheses [… 2 ml/kg iopamidol (Iopamiron®, 300 mg/ml, etc.)].

-       Lines 76-81. Was this procedure performed with the awake animal? Was sedation performed? This should be mentioned in the text. Doses and drugs should be reported.

-       Lines 85-87. The sentence has an awkward construct. Rephrase please.

-       Line 102. The title of the paragraph is “Treatment”, but only the surgical treatment is described. Didn’t the cat have other medications? It should be reported.

-       Line 121. Why were the edges not revived? In this instance, why an adsorbable suture was used? Wouldn’t it pose the risk of a recurrence after the suture absorption, which is quite rapid with this kind of suture? Moreover, there is not a follow-up for this case. A reader could perceive that this is a standard treatment for this condition. The reason for this choice should be mentioned here, and discussed later.

-       Line 126. I would not title this section “evolution”, since this refers to the first post-operative phase only. Was there a follow-up for this case? The follow-up should be mentioned. This would be helpful to the reader, particularly for the considerations made for line 121.

-       Line 137 and following. I would avoid this subdivision in many paragraphs, as the main topic is the same. This subdivision makes the text redundant, as many concepts are present “here and there”, without a fluent text.

-       Lines 138-139. Same as for line 15.

-       Lines 146-149. This is “over discussion”, could be deleted.

-       Line 191. Why was it attributed wrongly to Dominique Jean Larrey? This should be mentioned, or the word deleted.

-       Lines 218-221. The sentence is not fluent, should be rephrased.

-       Line 220-221. “… represent at least 95% of congenital …”. This is already stated before. Should be removed.

-       Line 222. It is clear that this would be a caudo-cranial view during a laparotomy in dorsal recumbency. Nonetheless, it should be reported in the Figure description and landmarks could be added in the Figure.

-       Lines 282-283. Also indicate the indications, the contraindications, and the possible adverse effects of this technique, which likely explain its limited utilization.

-       Lines 283-288. Why was not a CT scan performed? The reason should be mentioned.

-       Line 289. The title is “Indications for surgery”, but here there is a discussion about the surgery, and not about the indications.

-       Line 291. “Prophylactic surgery”. This suggests that it is an elective surgery, which is unlikely to be indicated after an acute trauma, although there were no apparent severe damages. It is not clear whether time has elapsed from the fall to the surgery or if it was performed on the same day. In this case, why was the decision made to intervene immediately if this condition did not pose any problems, and why wait to fully stabilize the patient's condition?

-       Line 304. Before the Conclusions sections, there should be a part of the discussion regarding the possible limitations in the management of this patient.

-       Lines 314-316. Usually, the MDPI journals offer some standard attributes for the Author contributions. I believe, for example, that performing the surgery or having treated the cat is indicated by “Investigation”.

Minor:

-       Line 14. The word “any” could be removed.

-       Line 24. I believe that the “6” is a typo. Remove.

-       Line 25. Useful.

-       Line 29: please re-order the keywords alphabetically.

-       Line 30. Incidentaloma, with a single “L”.

-       Line 42. “Which occurred” could be removed.

-       Line 42-43. “There were no medical incidents, …”. A medical incident is an unintended or unexpected event that harmed a patient or caregiver. I would modify the phrase to “No other issues were reported in the medical history”.

-       Line 47. “Breaths” instead of “respiration”.

-       Lines 50-51. “confirming the occurrence of trauma”. I would remove this last part. Moreover, trauma was already reported in the anamnesis.

-       Line 55. Usual > range

-       Line 56. I would add “…(156 mmHg), which was attributed to stress or pain”.

-       Line 56. mm Hg > mmHg

-       Lines 57-62. Split the phrase in two sentences.

-       Lines 63-64. “of the liver or other”, remove.

-       Line 65. “however, the mass was notably circumscribed”. Remove, as it was already written before.

-       Line 132. “…kidneys, as it was …”

-       Line 161. “Pericardoperitoneal” is written here, while at line 72 we find “peritoneopericardial”. Choose one form to use.

-       Line 181. “Vosges et al. [11] and Carriou et al. [15].

-       Line 210. Note > note

-       Line 211. Delete one space after crus.

-       Line 211. Is > was

-       Line 217. As far as we are aware > based on a literature search

-       Line 270. Was already not in doubt > was not supported

I hope this will improve your manuscript.

Kind regards

Author Response

Dear reviewer,

Thank you for your careful reading.

The case report submitted by Deschamps et al. provides potentially relevant information regarding true diaphragmatic hernia in cats. Its main strength lies in the positive outcome.

However, the report requires some improvements to be accepted for publication.

The Introduction section is quite short, thus does not give a good insight into the topic. Some parts of the Discussion section should be moved to the Introduction.

Corrected.

I have put the first paragraphs of the discussion in the introduction.

The Case presentation section lacks information about the overall treatments performed to the cat, the time elapsed between events and the choices underlying certain treatments.

The Discussion section is interesting and clear but it could be perceived as redundant in some parts, as most of the concepts are repeated in different sections. The subdivision in many paragraphs is not so useful: the text could be structured as a single unit and written more concisely and with less redundancy. In this form, it seem a monography about the topic of hernia, rather than a discussion of the case description.
In this section, a part regarding the possible limitations of this patient’s management is lacking. There is no mention about the contraindication and adverse effects of the contrast peritoneography.

Specific comments are listed below (please note that this refers to the Version 1, since this review was made on it. The version 2 was uploaded later. The added simple summary is overall good).

Below are the details of the corrections.

Major:

-       Line 2. Since the terminology “Morgagni hernia” has never been used in cats, as you report, it is hazardous to use it directly in the title. I suggest to leave the description in the Abstract and in the keywords, but to remove it from the title.

I am very keen on having the term 'Morgagni hernia' appear in the title because it's the best way to be cited when this type of hernia is described. It is also the way to express that this is officially the first case of Morgagni Hernia described in veterinary medicine which is the whole interest of this publication. There is no doubt that it corresponds well to the entity described under this name in human medicine.

-       Line 15. The word classic is used throughout the text (line 138, 229, and others), most of the times written within the quotation marks (“classic”). Personally, I would avoid using this term throughout the text to refer to a traumatic hernia. Instead, I would simply use “traumatic hernia”. This makes the text more fluid and formal. I recommend making similar adjustments in subsequent instances where the term “classic” is used.

Corrected

I have replaced 'classic' with 'traumatic' throughout the text.

-       Line 32. The introduction section is too short. Please extend it, maybe moving here some parts from the Discussion section.

Corrected.

I have put the first three paragraphs of the discussion in the introduction.

       Line 57. The presence of a mass is not confirmed, as this is only a radiographic interpretation. Thus, using the term “mass” is not correct in this phase. I would modify to “radiodense area”. I would change it throughout the text (line 77, and others).

Corrected.

I used the term “radiodense area”.

-       Line 75. “front view (dorso-ventral)”. As the cat did not exhibited dyspnea, why a dorso-ventral projection was chosen, instead of a ventro-dorsal? Or is it just an error in the text? The same for the next Figures (lines 91, 135).

It's a mistake because we perform all our radiographic images in a ventro-dorsal position. Thank you for your vigilance. I have corrected it everywhere.

-       Lines 76-89. Were all these procedures and the surgery performed on the same day?

The animal was received at night. Peritoneography was performed the next morning and surgery followed immediately, as seen on the x-ray: the contrast agent is visible in the bladder on the postoperative radiograph. I specified: 'The following day'... and 'In the wake of peritoneography'...

-       Line 79. Insert the drug name followed by the manifacturer’s information within parentheses [… 2 ml/kg iopamidol (Iopamiron®, 300 mg/ml, etc.)].

Corrected :

The following day, a positive contrast peritoneography was carried out without sedation by injecting transabdominally, on the linea alba, 2 ml/kg of an iodine derivative, iopamidol (Iopamiron®, 300 mg/ml), at room temperature and then placing the animal in sternal recumbency with the pelvis elevated (Figures 2a and 2b).

-       Lines 76-81. Was this procedure performed with the awake animal? Was sedation performed? This should be mentioned in the text. Doses and drugs should be reported.

Corrected :

The following day, a positive contrast peritoneography was carried out without sedation by injecting transabdominally, …

The cat was anesthetized with an association of diazepam (Diazépam TVM®, 0.25 mg/kg IV), propofol (Propovet®, 4 mg/kg IV) and morphine (Morphine Aguettant®; 0.05 mg/kg) then maintained with isoflurane (Vetflurane®, 1.5%).

-       Lines 85-87. The sentence has an awkward construct. Rephrase please.

Corrected:

“The injected contrast agent, now present at the site of the abnormal structure previously seen in plain radiographs, confirmed the abdominal origin of this structure.”

-       Line 102. The title of the paragraph is “Treatment”, but only the surgical treatment is described. Didn’t the cat have other medications? It should be reported.

Corrected :

I used the term ‘surgical treatment’ ; there was no medical treatment.

-       Line 121. Why were the edges not revived? In this instance, why an adsorbable suture was used? Wouldn’t it pose the risk of a recurrence after the suture absorption, which is quite rapid with this kind of suture? Moreover, there is not a follow-up for this case. A reader could perceive that this is a standard treatment for this condition. The reason for this choice should be mentioned here, and discussed later.

The surgical procedure for diaphragmatic hernias does not advise refreshing the edges of the diaphragmatic wound and recommends the use of absorbable sutures. The use of non-absorbable sutures is not recommended. It is the healing at the suture points that allows for the adhesion of the diaphragm's edges. A braided suture is more inflammatory than a monofilament, which is why we prefer Vicryl®. These recommendations are consistent with those found in surgical textbooks and authors mentioned in the references (notably Rose).

-       Line 126. I would not title this section “evolution”, since this refers to the first post-operative phase only. Was there a follow-up for this case? The follow-up should be mentioned. This would be helpful to the reader, particularly for the considerations made for line 121.

Corrected:

I added: “The postoperative follow-up confirmed the healing of all lesions.”

-       Line 137 and following. I would avoid this subdivision in many paragraphs, as the main topic is the same. This subdivision makes the text redundant, as many concepts are present “here and there”, without a fluent text.

I have deleted some subtitles as requested but I have kept some so that the discussion does not appear as an indigestible block.

-       Lines 138-139. Same as for line 15.

Corrected

-       Lines 146-149. This is “over discussion”, could be deleted.

I deleted the sentence: “Similarly, it is not accurate to refer to a traumatic “abdominal hernia” when organs protrude through a breach in the muscular wall of the abdomen and are palpable under the skin: without a hernial sac, strictly speaking, there is no hernia; the term “evisceration” is more appropriate”.

-       Line 191. Why was it attributed wrongly to Dominique Jean Larrey? This should be mentioned, or the word deleted.

I deleted “wrongly”.

It's a pity to perpetuate an error of attribution but I find it would be off-topic and too lengthy to elaborate on the historical controversy. This simple adjective specified the wrong attribution without making the text heavy.

-       Lines 218-221. The sentence is not fluent, should be rephrased.

I rephrased :

“In cats, all true diaphragmatic hernias are ventral, either right or left;  there have been no reports of hernias that correspond to the Bochdalek hernia seen in humans (posterolateral) (Figures 4). In contrast, in humans, Morgagni and Larrey hernias account for only 2 to 5% of congenital diaphragmatic hernias and Bochdalek hernias represent at least 95% of congenital diaphragmatic hernias [31,32].”

-       Line 220-221. “… represent at least 95% of congenital …”. This is already stated before. Should be removed.

The previous sentence was deleted: “Indeed, Bochdalek hernias represent 95% of congenital diaphragmatic hernias in humans, but”.

-       Line 222. It is clear that this would be a caudo-cranial view during a laparotomy in dorsal recumbency. Nonetheless, it should be reported in the Figure description and landmarks could be added in the Figure.

Done.

Figure 4: Location of diaphragmatic hernias in cats (view from the back, cat on its back).

-       Lines 282-283. Also indicate the indications, the contraindications, and the possible adverse effects of this technique, which likely explain its limited utilization.

I added:

“Iopamidol, a nonionic water-soluble contrast medium, has been evaluated for its reactivity when used in peritoneal contexts, like peritoneography; it does not provoke significant peritoneal reactivity [46].”

-       Lines 283-288. Why was not a CT scan performed? The reason should be mentioned.

I wrote:

“Peritoneography is advantageous for its simplicity, lack of sedation requirement, and accessibility to all veterinarians.“

If this examination had not been conclusive, we would have performed a scan.

I corrected : “Peritoneography is advantageous for its simplicity, lack of sedation requirement, and accessibility to all veterinarians that's why it was chosen as the first option.”

-       Line 289. The title is “Indications for surgery”, but here there is a discussion about the surgery, and not about the indications.

Corrected.

I deleted “Indications for”

However, the paragraph discusses the relevance of the surgery (not the surgical techniques), therefore the indications.

-       Line 291. “Prophylactic surgery”. This suggests that it is an elective surgery, which is unlikely to be indicated after an acute trauma, although there were no apparent severe damages. It is not clear whether time has elapsed from the fall to the surgery or if it was performed on the same day. In this case, why was the decision made to intervene immediately if this condition did not pose any problems, and why wait to fully stabilize the patient's condition?

There was no reason to wait to stabilize a patient who was not unstable: he had no symptoms. In the absence of symptoms, there was no benefit to delaying surgery. Since the patient was present, the surgery was carried out immediately following the diagnosis, out of concern for efficiency. The surgery was done in the wake of the peritoneography. I removed the word 'prophylactic' to satisfy the reviewer, but it was nevertheless a prophylactic surgery, in anticipation of organ entrapment.

-       Line 304. Before the Conclusions sections, there should be a part of the discussion regarding the possible limitations in the management of this patient.

I made this discussion:

“However, the decision for surgery in this particular case could be seen as debatable since a true diaphragmatic hernia was confirmed via peritoneography, with the organs theoretically contained within a hernial sac and, thus, unlikely to massively protrude into the thoracic cavity.”

-       Lines 314-316. Usually, the MDPI journals offer some standard attributes for the Author contributions. I believe, for example, that performing the surgery or having treated the cat is indicated by “Investigation”.

Corrected

Minor:

-       Line 14. The word “any” could be removed.

Corrected

-       Line 24. I believe that the “6” is a typo. Remove.

It’s not a typo. I wrote “six” in full letters.

-       Line 25. Useful.

Corrected

-       Line 29: please re-order the keywords alphabetically.

I do not believe that the keywords need to be put in alphabetical order; I think they should be placed in order of relevance. I let the editor make corrections if needed.

-       Line 30. Incidentaloma, with a single “L”.

Corrected

-       Line 42. “Which occurred” could be removed.

Corrected

-       Line 42-43. “There were no medical incidents, …”. A medical incident is an unintended or unexpected event that harmed a patient or caregiver. I would modify the phrase to “No other issues were reported in the medical history”.

Corrected:

“No other issues, including trauma, were reported in the medical history prior to this event.”

-       Line 47. “Breaths” instead of “respiration”.

Corrected

-       Lines 50-51. “confirming the occurrence of trauma”. I would remove this last part. Moreover, trauma was already reported in the anamnesis.

Corrected

-       Line 55. Usual > range

Corrected

-       Line 56. I would add “…(156 mmHg), which was attributed to stress or pain”.

Corrected

-       Line 56. mm Hg > mmHg

Corrected

-       Lines 57-62. Split the phrase in two sentences.

Corrected

-       Lines 63-64. “of the liver or other”, remove.

Corrected

-       Line 65. “however, the mass was notably circumscribed”. Remove, as it was already written before.

Corrected

-       Line 132. “…kidneys, as it was …”

Corrected

-       Line 161. “Pericardoperitoneal” is written here, while at line 72 we find “peritoneopericardial”. Choose one form to use.

Corrected

-       Line 181. “Vosges et al. [11] and Carriou et al. [15].

Corrected

-       Line 210. Note > note

I’m not sure to understand because you wrote “note”. I assumed you meant “noted”

I replaced “note” by “noted”.

-       Line 211. Delete one space after crus.

Corrected

-       Line 211. Is > was

Corrected

-       Line 217. As far as we are aware > based on a literature search

Corrected.

-       Line 270. Was already not in doubt > was not supported

Corrected

I hope this will improve your manuscript.

Yes, it will.

Thank you very much for all the time you have spent on the very careful proofreading of this article. Thank you for the improvements made. I hope my corrections are to your satisfaction.

Best regards

Reviewer 4 Report

Comments and Suggestions for Authors

thank you for an interesting paper. I do have a couple of questions.

1. line 56 is this systolic blood pressure

2. line 79 the constrast, iopamidol, has no tissue irritation or contraindications to its use?

3. Line 129 the hernial sac is left in place after closing the defect, does it remain static or  slowly degenerate?

4. Figure 4, what is the orientation for this picture, would the animal be standing or lying on its back?

Author Response

Thank you very much for the time you have devoted to reviewing this article and for the favorable impressions you have expressed.

I do have a couple of questions.

  1. line 56 is this systolic blood pressure ?

Yes it is.

Line 62: A systolic blood pressure measurement revealed slight hypertension (156 mm Hg) attributed to stress or pain.

  1. line 79 the constrast, iopamidol, has no tissue irritation or contraindications to its use?

Iopamidol, a nonionic water-soluble contrast medium, has been evaluated for its reactivity when used in peritoneal contexts, like peritoneography. Research indicates that Iopamidol does not provoke significant peritoneal reactivity, making it a potentially suitable contrast agent for evaluating patients with possible bowel perforation. A comparative study involving intraperitoneal injections of ionic and nonionic contrast agents in rats found that only intraperitoneal barium injection produced significant tissue reactions, such as adhesions and ascites, while no difference was observed with Iopamidol compared to other water-soluble contrast agents. This suggests that Iopamidol might satisfy the need for nonreactive and nearly iso-osmolar contrast agents for such diagnostic purposes, although its higher cost may limit clinical application.

Invest Radiol. 1990 Feb;25(2):141-5.

Iopamidol as a gastrointestinal contrast agent. Lack of peritoneal reactivity

S L Ferrante 1, J S Schreiman, J W Rouse, J A Rysavy, S C Cheng, M P Frick

DOI: 10.1097/00004424-199002000-00008

Additionally, the compatibility and stability of non-ionic iodinated contrast agents like Iopamidol when mixed with peritoneal dialysis solution for CT peritoneography have been studied. The findings indicate that Iopamidol, along with other non-ionic iodinated contrast agents, remains chemically stable for up to five days under various storage conditions. The mixture did not show any changes in visual appearance, turbidity, pH, drug concentration, or chemical degradation, making these agents safe for use in CT peritoneography protocols. This is significant for ensuring patient safety and the diagnostic reliability of imaging studies.

Perit Dial Int. 2023 Mar;43(2):151-158.

Compatibility and stability of non-ionic iodinated contrast media in peritoneal dialysis solution and safe practice considerations for CT peritoneography

Jayan Rappai 1, John H Crabtree 2, Ann Mancini 3, Sudheer Kumar Badugu 1, Anuj Kaushal 1, Mary E Gellens 3

DOI: 10.1177/08968608221096562

Based on these findings, Iopamidol appears to have minimal tissue irritation risk and shows no significant contraindications for peritoneography, given its stability and lack of reactivity in peritoneal use.

  1. Line 129 the hernial sac is left in place after closing the defect, does it remain static or  slowly degenerate?

Since the vascularization of the hernial sac is not altered by the surgery, it is reasonable to assume that the sac remains static.

  1. Figure 4, what is the orientation for this picture, would the animal be standing or lying on its back?

I added:

Figure 4: Location of diaphragmatic hernias in cats (view from the back, cat on its back).

Thank you again for your time.

Round 2

Reviewer 1 Report

Comments and Suggestions for Authors

The anaesthetic protocol used should be added to the case description.

I do not fully agree with the methodology used when you say that you did not reviving the edges of the diaphragm. Contrary, I agree with your decision to use the hernia sac for hernia reduction, I suggest you include references to this: 

https://doi.org/10.3390/ani12233240

Spadola F., Costa G.L., Morici M., Interlandi C., Nastasi B., Musicò M. (2017) Autologous prosthesis for the surgery of two simultaneous hernias in a calf. Large Animal Review 23:195-197.

Author Response

Dear reviewer,

Thank you for the time spent reviewing this article.

I read the article with great interest, the study is interesting and can enrich our knowledge of the development of diaphragmatic hernias. But some clarifications need to be added before it can be considered for publication.

Treatment

The anaesthetic protocol used should be added to the case description.

Corrected. I added:

“The cat was anesthetized with an association of diazepam (Diazépam TVM®, 0.25 mg/kg IV), propofol (Propovet®, 4 mg/kg IV) and morphine (Morphine Aguettant®; 0.05 mg/kg) then maintained with isoflurane (Vetflurane®, 1.5%).”

I do not fully agree with the methodology used when you say that you did not reviving the edges of the diaphragm. Contrary, I agree with your decision to use the hernia sac for hernia reduction, I suggest you include references to this:

https://doi.org/10.3390/ani12233240

Spadola F., Costa G.L., Morici M., Interlandi C., Nastasi B., Musicò M. (2017) Autologous prosthesis for the surgery of two simultaneous hernias in a calf. Large Animal Review 23:195-197.

The surgical approach for diaphragmatic hernia repair does not recommend debriding the wound margins. Healing at the suture sites promotes the adhesion of the diaphragm's edges. These guidelines align with those presented in authoritative surgical textbooks and by authors cited in the references, notably Rose. The situation is indeed different in the publications you cite because the hernia sac is huge and warrants excision. That was not the case here. I prefer not to quote the references because they pertain to another species that were treated differently. I had to limit the references concerning the cat ; it would seem incongruous to cite solitaries references regarding a calf or a swine treated differently. Thank you for your understanding.

Thank you again for the time spent reviewing this article.

Best regards.

Reviewer 3 Report

Comments and Suggestions for Authors

The case report submitted by Deschamps et al. provides potentially relevant information regarding true diaphragmatic hernia in cats. Its main strength lies in the positive outcome.

However, the report requires some improvements to be accepted for publication.

The Introduction section is quite short, thus does not give a good insight into the topic. Some parts of the Discussion section should be moved to the Introduction.

The Case presentation section lacks information about the overall treatments performed to the cat, the time elapsed between events and the choices underlying certain treatments.

The Discussion section is interesting and clear but it could be perceived as redundant in some parts, as most of the concepts are repeated in different sections. The subdivision in many paragraphs is not so useful: the text could be structured as a single unit and written more concisely and with less redundancy. In this form, it seem a monography about the topic of hernia, rather than a discussion of the case description.
In this section, a part regarding the possible limitations of this patient’s management is lacking. There is no mention about the contraindication and adverse effects of the contrast peritoneography.

Specific comments are listed below (please note that this refers to the Version 1, since this review was made on it. The version 2 was uploaded later. The added simple summary is overall good).

Major:

-       Line 2. Since the terminology “Morgagni hernia” has never been used in cats, as you report, it is hazardous to use it directly in the title. I suggest to leave the description in the Abstract and in the keywords, but to remove it from the title.

-       Line 15. The word classic is used throughout the text (line 138, 229, and others), most of the times written within the quotation marks (“classic”). Personally, I would avoid using this term throughout the text to refer to a traumatic hernia. Instead, I would simply use “traumatic hernia”. This makes the text more fluid and formal. I recommend making similar adjustments in subsequent instances where the term “classic” is used.

-       Line 32. The introduction section is too short. Please extend it, maybe moving here some parts from the Discussion section.

-       Line 57. The presence of a mass is not confirmed, as this is only a radiographic interpretation. Thus, using the term “mass” is not correct in this phase. I would modify to “radiodense area”. I would change it throughout the text (line 77, and others).

-       Line 75. “front view (dorso-ventral)”. As the cat did not exhibited dyspnea, why a dorso-ventral projection was chosen, instead of a ventro-dorsal? Or is it just an error in the text? The same for the next Figures (lines 91, 135).

-       Lines 76-89. Were all these procedures and the surgery performed on the same day?

-       Line 79. Insert the drug name followed by the manifacturer’s information within parentheses [… 2 ml/kg iopamidol (Iopamiron®, 300 mg/ml, etc.)].

-       Lines 76-81. Was this procedure performed with the awake animal? Was sedation performed? This should be mentioned in the text. Doses and drugs should be reported.

-       Lines 85-87. The sentence has an awkward construct. Rephrase please.

-       Line 102. The title of the paragraph is “Treatment”, but only the surgical treatment is described. Didn’t the cat have other medications? It should be reported.

-       Line 121. Why were the edges not revived? In this instance, why an adsorbable suture was used? Wouldn’t it pose the risk of a recurrence after the suture absorption, which is quite rapid with this kind of suture? Moreover, there is not a follow-up for this case. A reader could perceive that this is a standard treatment for this condition. The reason for this choice should be mentioned here, and discussed later.

-       Line 126. I would not title this section “evolution”, since this refers to the first post-operative phase only. Was there a follow-up for this case? The follow-up should be mentioned. This would be helpful to the reader, particularly for the considerations made for line 121.

-       Line 137 and following. I would avoid this subdivision in many paragraphs, as the main topic is the same. This subdivision makes the text redundant, as many concepts are present “here and there”, without a fluent text.

-       Lines 138-139. Same as for line 15.

-       Lines 146-149. This is “over discussion”, could be deleted.

-       Line 191. Why was it attributed wrongly to Dominique Jean Larrey? This should be mentioned, or the word deleted.

-       Lines 218-221. The sentence is not fluent, should be rephrased.

-       Line 220-221. “… represent at least 95% of congenital …”. This is already stated before. Should be removed.

-       Line 222. It is clear that this would be a caudo-cranial view during a laparotomy in dorsal recumbency. Nonetheless, it should be reported in the Figure description and landmarks could be added in the Figure.

-       Lines 282-283. Also indicate the indications, the contraindications, and the possible adverse effects of this technique, which likely explain its limited utilization.

-       Lines 283-288. Why was not a CT scan performed? The reason should be mentioned.

-       Line 289. The title is “Indications for surgery”, but here there is a discussion about the surgery, and not about the indications.

-       Line 291. “Prophylactic surgery”. This suggests that it is an elective surgery, which is unlikely to be indicated after an acute trauma, although there were no apparent severe damages. It is not clear whether time has elapsed from the fall to the surgery or if it was performed on the same day. In this case, why was the decision made to intervene immediately if this condition did not pose any problems, and why wait to fully stabilize the patient's condition?

-       Line 304. Before the Conclusions sections, there should be a part of the discussion regarding the possible limitations in the management of this patient.

-       Lines 314-316. Usually, the MDPI journals offer some standard attributes for the Author contributions. I believe, for example, that performing the surgery or having treated the cat is indicated by “Investigation”.

Minor:

-       Line 14. The word “any” could be removed.

-       Line 24. I believe that the “6” is a typo. Remove.

-       Line 25. Useful.

-       Line 29: please re-order the keywords alphabetically.

-       Line 30. Incidentaloma, with a single “L”.

-       Line 42. “Which occurred” could be removed.

-       Line 42-43. “There were no medical incidents, …”. A medical incident is an unintended or unexpected event that harmed a patient or caregiver. I would modify the phrase to “No other issues were reported in the medical history”.

-       Line 47. “Breaths” instead of “respiration”.

-       Lines 50-51. “confirming the occurrence of trauma”. I would remove this last part. Moreover, trauma was already reported in the anamnesis.

-       Line 55. Usual > range

-       Line 56. I would add “…(156 mmHg), which was attributed to stress or pain”.

-       Line 56. mm Hg > mmHg

-       Lines 57-62. Split the phrase in two sentences.

-       Lines 63-64. “of the liver or other”, remove.

-       Line 65. “however, the mass was notably circumscribed”. Remove, as it was already written before.

-       Line 132. “…kidneys, as it was …”

-       Line 161. “Pericardoperitoneal” is written here, while at line 72 we find “peritoneopericardial”. Choose one form to use.

-       Line 181. “Vosges et al. [11] and Carriou et al. [15].

-       Line 210. Note > note

-       Line 211. Delete one space after crus.

-       Line 211. Is > was

-       Line 217. As far as we are aware > based on a literature search

-       Line 270. Was already not in doubt > was not supported

I hope this will improve your manuscript.

Kind regards

Author Response

Dear reviewer,

Thank you for your careful reading.

The case report submitted by Deschamps et al. provides potentially relevant information regarding true diaphragmatic hernia in cats. Its main strength lies in the positive outcome.

However, the report requires some improvements to be accepted for publication.

The Introduction section is quite short, thus does not give a good insight into the topic. Some parts of the Discussion section should be moved to the Introduction.

Corrected.

I have put the first paragraphs of the discussion in the introduction.

The Case presentation section lacks information about the overall treatments performed to the cat, the time elapsed between events and the choices underlying certain treatments.

The Discussion section is interesting and clear but it could be perceived as redundant in some parts, as most of the concepts are repeated in different sections. The subdivision in many paragraphs is not so useful: the text could be structured as a single unit and written more concisely and with less redundancy. In this form, it seem a monography about the topic of hernia, rather than a discussion of the case description.
In this section, a part regarding the possible limitations of this patient’s management is lacking. There is no mention about the contraindication and adverse effects of the contrast peritoneography.

Specific comments are listed below (please note that this refers to the Version 1, since this review was made on it. The version 2 was uploaded later. The added simple summary is overall good).

Below are the details of the corrections.

Major:

-       Line 2. Since the terminology “Morgagni hernia” has never been used in cats, as you report, it is hazardous to use it directly in the title. I suggest to leave the description in the Abstract and in the keywords, but to remove it from the title.

I am very keen on having the term 'Morgagni hernia' appear in the title because it's the best way to be cited when this type of hernia is described. It is also the way to express that this is officially the first case of Morgagni Hernia described in veterinary medicine which is the whole interest of this publication. There is no doubt that it corresponds well to the entity described under this name in human medicine.

-       Line 15. The word classic is used throughout the text (line 138, 229, and others), most of the times written within the quotation marks (“classic”). Personally, I would avoid using this term throughout the text to refer to a traumatic hernia. Instead, I would simply use “traumatic hernia”. This makes the text more fluid and formal. I recommend making similar adjustments in subsequent instances where the term “classic” is used.

Corrected

I have replaced 'classic' with 'traumatic' throughout the text.

-       Line 32. The introduction section is too short. Please extend it, maybe moving here some parts from the Discussion section.

Corrected.

I have put the first three paragraphs of the discussion in the introduction.

-       Line 57. The presence of a mass is not confirmed, as this is only a radiographic interpretation. Thus, using the term “mass” is not correct in this phase. I would modify to “radiodense area”. I would change it throughout the text (line 77, and others).

Corrected.

I used the term “radiodense area”.

-       Line 75. “front view (dorso-ventral)”. As the cat did not exhibited dyspnea, why a dorso-ventral projection was chosen, instead of a ventro-dorsal? Or is it just an error in the text? The same for the next Figures (lines 91, 135).

It's a mistake because we perform all our radiographic images in a ventro-dorsal position. Thank you for your vigilance. I have corrected it everywhere.

-       Lines 76-89. Were all these procedures and the surgery performed on the same day?

The animal was received at night. Peritoneography was performed the next morning and surgery followed immediately, as seen on the x-ray: the contrast agent is visible in the bladder on the postoperative radiograph. I specified: 'The following day'... and 'In the wake of peritoneography'...

-       Line 79. Insert the drug name followed by the manifacturer’s information within parentheses [… 2 ml/kg iopamidol (Iopamiron®, 300 mg/ml, etc.)].

Corrected :

The following day, a positive contrast peritoneography was carried out without sedation by injecting transabdominally, on the linea alba, 2 ml/kg of an iodine derivative, iopamidol (Iopamiron®, 300 mg/ml), at room temperature and then placing the animal in sternal recumbency with the pelvis elevated (Figures 2a and 2b).

-       Lines 76-81. Was this procedure performed with the awake animal? Was sedation performed? This should be mentioned in the text. Doses and drugs should be reported.

Corrected :

The following day, a positive contrast peritoneography was carried out without sedation by injecting transabdominally, …

The cat was anesthetized with an association of diazepam (Diazépam TVM®, 0.25 mg/kg IV), propofol (Propovet®, 4 mg/kg IV) and morphine (Morphine Aguettant®; 0.05 mg/kg) then maintained with isoflurane (Vetflurane®, 1.5%).

-       Lines 85-87. The sentence has an awkward construct. Rephrase please.

Corrected:

“The injected contrast agent, now present at the site of the abnormal structure previously seen in plain radiographs, confirmed the abdominal origin of this structure.”

-       Line 102. The title of the paragraph is “Treatment”, but only the surgical treatment is described. Didn’t the cat have other medications? It should be reported.

Corrected :

I used the term ‘surgical treatment’ ; there was no medical treatment.

-       Line 121. Why were the edges not revived? In this instance, why an adsorbable suture was used? Wouldn’t it pose the risk of a recurrence after the suture absorption, which is quite rapid with this kind of suture? Moreover, there is not a follow-up for this case. A reader could perceive that this is a standard treatment for this condition. The reason for this choice should be mentioned here, and discussed later.

The surgical procedure for diaphragmatic hernias does not advise refreshing the edges of the diaphragmatic wound and recommends the use of absorbable sutures. The use of non-absorbable sutures is not recommended. It is the healing at the suture points that allows for the adhesion of the diaphragm's edges. A braided suture is more inflammatory than a monofilament, which is why we prefer Vicryl®. These recommendations are consistent with those found in surgical textbooks and authors mentioned in the references (notably Rose).

-       Line 126. I would not title this section “evolution”, since this refers to the first post-operative phase only. Was there a follow-up for this case? The follow-up should be mentioned. This would be helpful to the reader, particularly for the considerations made for line 121.

Corrected:

I added: “The postoperative follow-up confirmed the healing of all lesions.”

-       Line 137 and following. I would avoid this subdivision in many paragraphs, as the main topic is the same. This subdivision makes the text redundant, as many concepts are present “here and there”, without a fluent text.

I have deleted some subtitles as requested but I have kept some so that the discussion does not appear as an indigestible block.

-       Lines 138-139. Same as for line 15.

Corrected

-       Lines 146-149. This is “over discussion”, could be deleted.

I deleted the sentence: “Similarly, it is not accurate to refer to a traumatic “abdominal hernia” when organs protrude through a breach in the muscular wall of the abdomen and are palpable under the skin: without a hernial sac, strictly speaking, there is no hernia; the term “evisceration” is more appropriate”.

-       Line 191. Why was it attributed wrongly to Dominique Jean Larrey? This should be mentioned, or the word deleted.

I deleted “wrongly”.

It's a pity to perpetuate an error of attribution but I find it would be off-topic and too lengthy to elaborate on the historical controversy. This simple adjective specified the wrong attribution without making the text heavy.

-       Lines 218-221. The sentence is not fluent, should be rephrased.

I rephrased :

“In cats, all true diaphragmatic hernias are ventral, either right or left;  there have been no reports of hernias that correspond to the Bochdalek hernia seen in humans (posterolateral) (Figures 4). In contrast, in humans, Morgagni and Larrey hernias account for only 2 to 5% of congenital diaphragmatic hernias and Bochdalek hernias represent at least 95% of congenital diaphragmatic hernias [31,32].”

-       Line 220-221. “… represent at least 95% of congenital …”. This is already stated before. Should be removed.

The previous sentence was deleted: “Indeed, Bochdalek hernias represent 95% of congenital diaphragmatic hernias in humans, but”.

-       Line 222. It is clear that this would be a caudo-cranial view during a laparotomy in dorsal recumbency. Nonetheless, it should be reported in the Figure description and landmarks could be added in the Figure.

Done.

Figure 4: Location of diaphragmatic hernias in cats (view from the back, cat on its back).

-       Lines 282-283. Also indicate the indications, the contraindications, and the possible adverse effects of this technique, which likely explain its limited utilization.

I added:

“Iopamidol, a nonionic water-soluble contrast medium, has been evaluated for its reactivity when used in peritoneal contexts, like peritoneography; it does not provoke significant peritoneal reactivity [46].”

-       Lines 283-288. Why was not a CT scan performed? The reason should be mentioned.

I wrote:

“Peritoneography is advantageous for its simplicity, lack of sedation requirement, and accessibility to all veterinarians.“

If this examination had not been conclusive, we would have performed a scan.

I corrected : “Peritoneography is advantageous for its simplicity, lack of sedation requirement, and accessibility to all veterinarians that's why it was chosen as the first option.”

-       Line 289. The title is “Indications for surgery”, but here there is a discussion about the surgery, and not about the indications.

Corrected.

I deleted “Indications for”

However, the paragraph discusses the relevance of the surgery (not the surgical techniques), therefore the indications.

-       Line 291. “Prophylactic surgery”. This suggests that it is an elective surgery, which is unlikely to be indicated after an acute trauma, although there were no apparent severe damages. It is not clear whether time has elapsed from the fall to the surgery or if it was performed on the same day. In this case, why was the decision made to intervene immediately if this condition did not pose any problems, and why wait to fully stabilize the patient's condition?

There was no reason to wait to stabilize a patient who was not unstable: he had no symptoms. In the absence of symptoms, there was no benefit to delaying surgery. Since the patient was present, the surgery was carried out immediately following the diagnosis, out of concern for efficiency. The surgery was done in the wake of the peritoneography. I removed the word 'prophylactic' to satisfy the reviewer, but it was nevertheless a prophylactic surgery, in anticipation of organ entrapment.

-       Line 304. Before the Conclusions sections, there should be a part of the discussion regarding the possible limitations in the management of this patient.

I made this discussion:

“However, the decision for surgery in this particular case could be seen as debatable since a true diaphragmatic hernia was confirmed via peritoneography, with the organs theoretically contained within a hernial sac and, thus, unlikely to massively protrude into the thoracic cavity.”

-       Lines 314-316. Usually, the MDPI journals offer some standard attributes for the Author contributions. I believe, for example, that performing the surgery or having treated the cat is indicated by “Investigation”.

Corrected

Minor:

-       Line 14. The word “any” could be removed.

Corrected

-       Line 24. I believe that the “6” is a typo. Remove.

It’s not a typo. I wrote “six” in full letters.

-       Line 25. Useful.

Corrected

-       Line 29: please re-order the keywords alphabetically.

I do not believe that the keywords need to be put in alphabetical order; I think they should be placed in order of relevance. I let the editor make corrections if needed.

-       Line 30. Incidentaloma, with a single “L”.

Corrected

-       Line 42. “Which occurred” could be removed.

Corrected

-       Line 42-43. “There were no medical incidents, …”. A medical incident is an unintended or unexpected event that harmed a patient or caregiver. I would modify the phrase to “No other issues were reported in the medical history”.

Corrected:

“No other issues, including trauma, were reported in the medical history prior to this event.”

-       Line 47. “Breaths” instead of “respiration”.

Corrected

-       Lines 50-51. “confirming the occurrence of trauma”. I would remove this last part. Moreover, trauma was already reported in the anamnesis.

Corrected

-       Line 55. Usual > range

Corrected

-       Line 56. I would add “…(156 mmHg), which was attributed to stress or pain”.

Corrected

-       Line 56. mm Hg > mmHg

Corrected

-       Lines 57-62. Split the phrase in two sentences.

Corrected

-       Lines 63-64. “of the liver or other”, remove.

Corrected

-       Line 65. “however, the mass was notably circumscribed”. Remove, as it was already written before.

Corrected

-       Line 132. “…kidneys, as it was …”

Corrected

-       Line 161. “Pericardoperitoneal” is written here, while at line 72 we find “peritoneopericardial”. Choose one form to use.

Corrected

-       Line 181. “Vosges et al. [11] and Carriou et al. [15].

Corrected

-       Line 210. Note > note

I’m not sure to understand because you wrote “note”. I assumed you meant “noted”

I replaced “note” by “noted”.

-       Line 211. Delete one space after crus.

Corrected

-       Line 211. Is > was

Corrected

-       Line 217. As far as we are aware > based on a literature search

Corrected.

-       Line 270. Was already not in doubt > was not supported

Corrected

I hope this will improve your manuscript.

Yes, it will.

Thank you very much for all the time you have spent on the very careful proofreading of this article. Thank you for the improvements made. I hope my corrections are to your satisfaction.

Best regards

Round 3

Reviewer 3 Report

Comments and Suggestions for Authors

Dear Authors,

I believe the manuscript has improved, but there are still some issues worthy of discussion. Below are the comments:

- Line 28. I would avoid first-person style and instead prefer an impersonal style. Therefore, I would change "we believe…" to an impersonal phrase.

- Line 137. Was morphine also administered intravenously?

- Line 152 (formerly line 121). Here you can find the previous comment and response:

“-       Line 121. Why were the edges not revived? In this instance, why an adsorbable suture was used? Wouldn’t it pose the risk of a recurrence after the suture absorption, which is quite rapid with this kind of suture? Moreover, there is not a follow-up for this case. A reader could perceive that this is a standard treatment for this condition. The reason for this choice should be mentioned here, and discussed later.

 The surgical procedure for diaphragmatic hernias does not advise refreshing the edges of the diaphragmatic wound and recommends the use of absorbable sutures. The use of non-absorbable sutures is not recommended. It is the healing at the suture points that allows for the adhesion of the diaphragm's edges. A braided suture is more inflammatory than a monofilament, which is why we prefer Vicryl®. These recommendations are consistent with those found in surgical textbooks and authors mentioned in the references (notably Rose).”

The concept we intended to discuss concerns the use of absorbable suture and the lack of edge revival in this type of congenital hernia. Since the hernia is present from birth, it is difficult to think that the simple apposition of two serous membranes by suturing can allow a definitive healing. Similarly, it is risky to assume that the micro-trauma caused by the needle passage alone can promote healing of the margins of two serous membranes in contact. Clearly, the concept is not valid for a traumatic hernia, where the margins are vascularized due to trauma and there is no longer the physiological presence of the serous layer on the margin of the diaphragm subjected to traumatic laceration. In this case, therefore, simple juxtaposition is sufficient. In the case of a congenital hernia, the apposition of non-revived margins may simply keep them close until the suture thread is completely absorbed, without developing any scar tissue or healing of the defect. Again, I believe this aspect needs to be discussed. Furthermore, the follow-up is not mentioned for this patient. In fact, although it is stated at Line 165 that "the postoperative follow-up confirmed the healing of all lesions.", this is not sufficient. A comment on this topic will be made later. Additionally, I consider it risky to take as an example the treatment performed in a case report similar to this (Rose et al. 2017), based on a single case."

- Line 165. Here you can find the previous comment and response:

“-       Line 126. I would not title this section “evolution”, since this refers to the first post-operative phase only. Was there a follow-up for this case? The follow-up should be mentioned. This would be helpful to the reader, particularly for the considerations made for line 121.

Corrected:

I added: “The postoperative follow-up confirmed the healing of all lesions.””

As mentioned above, this sentence is not comprehensive of a follow-up that could allow for an objective evaluation. It is not indicated what type of investigation was performed. It is not indicated how long after the surgery the checks were carried out, and whether these times are longer than the complete absorption time of the Vicryl thread.

- Line 225. I suggest "dorsal recumbency" instead of "cat on its back."

- Line 286. I would delete "to all veterinarians." Also, I would change "that’s way" with a more formal construct. Moreover, peritoneography was not "the first option" as it was consequent to ultrasound scan after x-rays. For example: "… and accessibility. For this reason, it was performed in the diagnostic phase."

- Line 317. Here you can find the previous comment and response:

“-       Lines 314-316. Usually, the MDPI journals offer some standard attributes for the Author contributions. I believe, for example, that performing the surgery or having treated the cat is indicated by “Investigation”.

Corrected”

Veterinary Science clearly indicates the types of "Author contributions." Here they are:

[Author Contributions: For research articles with several authors, a short paragraph specifying their individual contributions must be provided. The following statements should be used 'Conceptualization, X.X. and Y.Y.; methodology, X.X.; software, X.X.; validation, X.X., Y.Y. and Z.Z.; formal analysis, X.X.; investigation, X.X.; resources, X.X.; data curation, X.X.; writing—original draft preparation, X.X.; writing—review and editing, X.X.; visualization, X.X.; supervision, X.X.; project administration, X.X.; funding acquisition, Y.Y. All authors have read and agreed to the published version of the manuscript.' Please turn to the CRediT taxonomy for the term explanation. Authorship must be limited to those who have contributed substantially to the work reported.]

Therefore, it is advisable to adhere to this style. I believe "performed the peritoneography and the surgery" and "took care of the cat" are not acceptable. Both, in my opinion, are encompassed by "Investigation."

Best regards

Author Response

Dear reviewer,

Thank you for your attentive second reading.

It's greatly appreciated to know that reviewers dedicate their time and skills to proofreading articles. As someone who often reviews articles myself, I am aware of the time it takes for people with busy schedules.

Thank you once more for your energy on our behalf.

First, thank you for allowing me to keep 'morgani hernia' in the title.

- Line 28. I would avoid first-person style and instead prefer an impersonal style. Therefore, I would change "we believe…" to an impersonal phrase.

Corrected.

I deleted “we believe that” and I added “probably”.

“Six other cases of Morgagni hernias have probably been reported in cats but were not identified as such.”

- Line 137. Was morphine also administered intravenously?

 Corrected.

I added “administered intravenously”.

- Line 152 (formerly line 121). Here you can find the previous comment and response:

“-       Line 121. Why were the edges not revived? In this instance, why an adsorbable suture was used? Wouldn’t it pose the risk of a recurrence after the suture absorption, which is quite rapid with this kind of suture? Moreover, there is not a follow-up for this case. A reader could perceive that this is a standard treatment for this condition. The reason for this choice should be mentioned here, and discussed later.

 The surgical procedure for diaphragmatic hernias does not advise refreshing the edges of the diaphragmatic wound and recommends the use of absorbable sutures. The use of non-absorbable sutures is not recommended. It is the healing at the suture points that allows for the adhesion of the diaphragm's edges. A braided suture is more inflammatory than a monofilament, which is why we prefer Vicryl®. These recommendations are consistent with those found in surgical textbooks and authors mentioned in the references (notably Rose).”

The concept we intended to discuss concerns the use of absorbable suture and the lack of edge revival in this type of congenital hernia. Since the hernia is present from birth, it is difficult to think that the simple apposition of two serous membranes by suturing can allow a definitive healing. Similarly, it is risky to assume that the micro-trauma caused by the needle passage alone can promote healing of the margins of two serous membranes in contact. Clearly, the concept is not valid for a traumatic hernia, where the margins are vascularized due to trauma and there is no longer the physiological presence of the serous layer on the margin of the diaphragm subjected to traumatic laceration. In this case, therefore, simple juxtaposition is sufficient. In the case of a congenital hernia, the apposition of non-revived margins may simply keep them close until the suture thread is completely absorbed, without developing any scar tissue or healing of the defect. Again, I believe this aspect needs to be discussed.

Furthermore, the follow-up is not mentioned for this patient. In fact, although it is stated at Line 165 that "the postoperative follow-up confirmed the healing of all lesions.", this is not sufficient. A comment on this topic will be made later. Additionally, I consider it risky to take as an example the treatment performed in a case report similar to this (Rose et al. 2017), based on a single case."

Dear reviewer,

I was ready to debate and argue about the procedure used but by presenting your remarks to the person who operated on the cat (Françoise Roux), she told me that she proceeded in the way you suggest! This is not how we usually proceed (for traumatic diaphragmatic hernias) which is why I had reported this procedure.

I now understand better why there was a postoperative pneumothorax.

I apologize for not having inquired about the precise technique she used and for having assumed that she operated as we are accustomed to. I also ask you to excuse her for not having carefully re-read this part of the manuscript.

I have therefore corrected the text accordingly:

« The edges of the diaphragm were revived by scissors section to promote healing of the margins. Herniorrhaphy was performed using a simple continuous closure with absorbable braided suture (Vicryl 3-0). The hernial sac was not removed. At the end of the procedure, transdiaphragmatic aspiration made it possible to collect several milliliters of air. »

I deleted the last sentence:

“Laparotomy can confirm the diagnosis, the operation is considered simple and low risk; due to the presence of a hernial sac, there is no open chest operating time ».

- Line 165. Here you can find the previous comment and response:

“-       Line 126. I would not title this section “evolution”, since this refers to the first post-operative phase only. Was there a follow-up for this case? The follow-up should be mentioned. This would be helpful to the reader, particularly for the considerations made for line 121.

Corrected:

I added: “The postoperative follow-up confirmed the healing of all lesions.””

As mentioned above, this sentence is not comprehensive of a follow-up that could allow for an objective evaluation. It is not indicated what type of investigation was performed. It is not indicated how long after the surgery the checks were carried out, and whether these times are longer than the complete absorption time of the Vicryl thread.

The cat was seen once for suture removal.

I've called these days to check in nearly one year to the day after the surgery, and all is well.

Corrected:

“The cat fully recovered and was discharged the day following surgery. Upon removal of the sutures, the cat was in perfect health: there was no dyspnea, the ranula had disappeared, and the skin lesions on the fingers were healed. Contacted by phone one year after the surgical intervention, the owners reported no health concerns. »

- Line 225. I suggest "dorsal recumbency" instead of "cat on its back."

 Corrected.

- Line 286. I would delete "to all veterinarians." Also, I would change "that’s way" with a more formal construct. Moreover, peritoneography was not "the first option" as it was consequent to ultrasound scan after x-rays. For example: "… and accessibility. For this reason, it was performed in the diagnostic phase."

I rephrased:

“Peritoneography has been selected due to its simplicity, the obviation of sedation, and its widespread accessibility to veterinarians.”

- Line 317. Here you can find the previous comment and response:

“-       Lines 314-316. Usually, the MDPI journals offer some standard attributes for the Author contributions. I believe, for example, that performing the surgery or having treated the cat is indicated by “Investigation”.

Corrected”

Veterinary Science clearly indicates the types of "Author contributions." Here they are:

[Author Contributions: For research articles with several authors, a short paragraph specifying their individual contributions must be provided. The following statements should be used 'Conceptualization, X.X. and Y.Y.; methodology, X.X.; software, X.X.; validation, X.X., Y.Y. and Z.Z.; formal analysis, X.X.; investigation, X.X.; resources, X.X.; data curation, X.X.; writing—original draft preparation, X.X.; writing—review and editing, X.X.; visualization, X.X.; supervision, X.X.; project administration, X.X.; funding acquisition, Y.Y. All authors have read and agreed to the published version of the manuscript.' Please turn to the CRediT taxonomy for the term explanation. Authorship must be limited to those who have contributed substantially to the work reported.]

Therefore, it is advisable to adhere to this style. I believe "performed the peritoneography and the surgery" and "took care of the cat" are not acceptable. Both, in my opinion, are encompassed by "Investigation."

Corrected.

Author Contributions: All authors have read and agreed to the published version of the manuscript. Investigation: F.A.R., T.C. and N.A.; writing—original draft preparation: J.-Y. D. and T. C.; writing—review and editing: J.-Y. D. and F.A.R.

I hope I have responded to your suggestions.

Best regards

Jack-Yves Deschamps

Thank you again, very sincerely: the manuscript is better.
